# Mechanism of *Curcuma wenyujin* Rhizoma on Acute Blood Stasis in Rats Based on a UPLC-Q/TOF-MS Metabolomics and Network Approach

**DOI:** 10.3390/molecules24010082

**Published:** 2018-12-27

**Authors:** Min Hao, De Ji, Lin Li, Lianlin Su, Wei Gu, Liya Gu, Qiaohan Wang, Tulin Lu, Chunqin Mao

**Affiliations:** School of Pharmacy, Nanjing University of Chinese Medicine, Nanjing 210023, China; hao_min0509@163.com (M.H.); jide3501@163.com (D.J.); lilin_med@163.com (L.L.); sulianlin1989@163.com (L.S.); guwei@njucm.edu.cn (W.G.); m13770513645_1@163.com (L.G.); wangqiaohan83@163.com (Q.W.)

**Keywords:** *Curcuma wenyujin* rhizome, UPLC-Q/TOF-MS, plasma metabolomics, multivariate statistical analysis, network approach

## Abstract

Rhizome of *Curcuma wenyujin*, which is called EZhu in China, is a traditional Chinese medicine used to treat blood stasis for many years. However, the underlying mechanism of EZhu is not clear at present. In this study, plasma metabolomics combined with network pharmacology were used to elucidate the therapeutic mechanism of EZhu in blood stasis from a metabolic perspective. The results showed that 26 potential metabolite markers of acute blood stasis were screened, and the levels were all reversed to different degrees by EZhu preadministration. Metabolic pathway analysis showed that the improvement of blood stasis by *Curcuma wenyujin* rhizome was mainly related to lipid metabolism (linoleic acid metabolism, ether lipid metabolism, sphingolipid metabolism, glycerophospholipid metabolism, and arachidonic acid metabolism) and amino acid metabolisms (tryptophan metabolism, lysine degradation). The component-target-pathway network showed that 68 target proteins were associated with 21 chemical components in EZhu. Five metabolic pathways of the network, including linoleic acid metabolism, sphingolipid metabolism, glycerolipid metabolism, arachidonic acid metabolism, and steroid hormone biosynthesis, were consistent with plasma metabolomics results. In conclusion, plasma metabolomics combined with network pharmacology can be helpful to clarify the mechanism of EZhu in improving blood stasis and to provide a literature basis for further research on the therapeutic mechanism of EZhu in clinical practice.

## 1. Introduction

Blood stasis syndrome (BSS), which results in retardation or cessation of blood flow, is widely known as Xueyu Zheng in China [1]. The theoretical basis of BSS can be traced back to “The Inner Canon of Huangdi” in the ancient pre-Qin period. Recent studies indicated that the possible aetiopathogenesis of BSS was relevant to the abnormality of hemorheology, including blood viscosity, blood sedimentation, erythrocyte aggregation, erythrocyte deformability, hematocrit, coagulant function, and microcirculation disturbance [2,3,4,5]. Some studies also showed that BSS is strongly associated with thrombogenesis, the inflammatory reaction, and edema hyperplasia and the immune response, which is induced by blood circulation disorders and the viscous state of systemic or local tissue organs [6]. The development of BSS can lead to many kinds of diseases, including hypertension, coronary heart disease, cerebral infarction, chronic pelvic infection, dysmenorrhoea, gastritis, arthritis, skin disease, and cancer [7,8,9,10,11]. In traditional Chinese medicine (TCM), Zheng is not merely a subclass disease, but also a type of common symptom discovery in different diseases. The doctors of TCM often recognize Zheng by identifying a little difference in the same symptoms of the same disease. Therefore, the distinction of Zheng makes TCM therapy become individualized in some degree [12]. In addition, patients with specific similar symptoms of different diseases could be treated by the same TCM treatment according to the theory of TCM, which has been accepted by the Consolidated Standards of Reporting Trials (CONSORT) for Chinese Herbal Medicine Formulas 2017 [13]. BSS is one of the common Zheng in TCM and has various symptoms and manifestations in clinical traditional Chinese medicine (TCM), such as distending pain or a tingling sensation in a fixed position, irritability or depression, dim complexion, lumps on the body, blood spots under the skin, unsmooth or string-like pulse, and purplish tongue or petechiae on the tongue [12,14]. The doctors of clinical Chinese medicine often use herbal medicine and Chinese medicine prescription to promote blood circulation and remove BSS for treatment, and this has achieved remarkable curative effects. Among these herbal medicines, *Curcuma wenyujin* rhizome, which is called EZhu in China, is a common herbal medicine to treat BSS [15]. It is widely used for various kinds of BSS in clinical TCM, for example, amenorrhea, dysmenorrhea, atherosclerosis, and other diseases, indicating remarkable curative effects. EZhu is rich in volatile oils consisting of monoterpenes and sesquiterpenes, besides slight curcuminoids. It was reported that β-elemene, curzerene, germacrone, curdione, neocurdione, and curcumenone were the dominant and bioactive ingredients [16,17]. However, the underlying mechanism of Ezhu is difficult to clarify because the multi-components and multiple targeting characteristics of Chinese herbal medicine play a common role in the curative effect.

In recent years, metabolomics technology has developed rapidly in the study of TCM, for its high adaptability in TCM study. Metabolomics is an unbiased analysis of the metabolites of the whole organism, which provides a powerful approach to discover biomarkers in biological systems and helps to promote the modernization of Chinese herbal medicine [18,19,20]. Furthermore, metabolomics pays close attention to the overall changes of all known and unknown molecular compounds, which is consistent with the overall view of TCM theory [21]. In this respect, the analysis of metabolite profiling of a disease model group and TCM treatment group in vivo could be helpful to study their clinical efficacy and mechanism of function. The primary techniques in metabolomics studies are gas chromatography-mass spectrometry (GC-MS) [22,23], liquid chromatography-mass spectrometry (LC-MS) [24,25,26], and nuclear magnetic resonance (NMR) spectroscopy [27,28] combined with multivariate statistical analysis. Among these techniques, LC-MS based methods are expected to be especially significant in metabolomics analyses, mainly due to the extensive availability of the technology and the compatibility with the extensive separation of biological samples [29,30,31]. However, single metabolomics research techniques still have their limitations. They only focus on changes in metabolites and cannot find the material basis for efficacy and effect targets. Network pharmacology can make up this defect very well in TCM study [32]. Network pharmacology is a new pattern of drug research to analyze the relationship between drugs, targets, metabolic pathways, and diseases by constructing a network model [33,34]. A drug-target-pathway-disease network for specific drugs, which was established on the existing data in the database, can help researchers in the analysis of the intervention and impact of drugs on the entire disease network and predict the effects of pharmacodynamic components on certain key targets and their pathways. The composition of TCM is diverse, and there are complex network relationships with diseases and targets. Therefore, metabolomics combined with network pharmacology is very suitable for the study of the action mechanism of TCM [35,36].

In this study, a ultra-performance liquid chromatography-quadrupole/time-of-flight mass spectrometry (UPLC-Q/TOF-MS) metabolomics method combined with network pharmacology was used to investigate the biochemical changes in BSS rats and the mechanism of EZhu. An acute BSS model in rats was established by twice subcutaneous adrenaline injection combined with an ice-water swim. Biochemical criterions, including whole blood viscosity, plasma viscosity, four coagulation functions (prothrombin time, PT; thrombin time, TT; activated partial thromboplastin time, APTT; fibrinogen, FIB), fibrinolytic factors (tissue-type plasminogen activator, t-PA; plasminogen activator inhibitor-1, PAI-1), and inflammatory factors (IL-6, TNF-α) were evaluated for the effects of EZhu on the model rats. The possible biomarkers related to the deranged metabolic pathways in BSS were identified, and a drug-target-pathway network was established to better understand BSS and the potential efficacies and mechanisms of EZhu in treating BSS.

## 2. Results

### 2.1. Hemorheology and Related Functional Parameters Evaluation

Hemorheology and coagulation functions are the main pharmacological indicators for blood stasis syndrome. In this study, the results of whole blood viscosity (WBV), including 1 s, 5 s, 30 s, and 200 s shear rates, plasma viscosity (PV), and coagulation function (APTT, TT, PT, FIB) are shown in Figure 1A–C. Compared with the NC group, WBV and PV in ABS group were significantly enhanced (*p* < 0.05). Compared with the ABS group, these parameters in the CST and EZ groups were significantly lowered (*p* < 0.05). These parameters showed that the ABS model was successfully established. In addition, both the CST and EZ groups had therapeutic effects on acute blood stasis syndrome. The values of APTT, TT, PT, and FIB in the ABS group were significantly longer (*p* < 0.01) compared with the NC group. The time of PT in the CST group had no significant difference with the NC group. Moreover, coagulation function in the CST and EZ groups was significantly decreased (*p* < 0.05) compared with the NC group.

In addition to hemorheology and coagulation function, the effects of the fibrinolytic system (t-PA, PAI-1) and inflammatory factors (IL-6, TNF-α) on the ABS model were also examined in this study. The results are shown in Figure 1D,E. Compared with the NC group, the value of t-PA was significantly lower (*p* < 0.05) than the ABS group, and the parameter was significantly improved (*p* < 0.01) in the CST and EZ groups. The parameters of PAI-1, IL-6, and TNF-α in the ABS group were significantly increased (*p* < 0.01) than that in the NC group, and was significantly improved (*p* < 0.05) after CST and EZ preadministration. The results manifested that ABS syndrome also had a great relationship with the fibrinolytic system and inflammation.

### 2.2. Plasma Metabolomics Analysis

The different groups of total ion chromatograms (TIC) in both positive and negative modes are shown in Figure 2A,B. The TICs were quite different from the positive to negative mode. However, the TICs in the same mode of different groups were not significantly different, indicating that the difference of endogenous metabolites in each group could not be seen directly from the TICs. Therefore, to further discover the differences among endogenous metabolites in each group, multivariate statistical analysis, including unsupervised principal component analysis (PCA) and supervised orthogonal partial least squares discriminant analysis (OPLS-DA) methods, was managed by Simca-P 14.1 (Umetrics, Umea, Sweden). The PCA score plots of plasma metabolic profiling of NC, ABS, and EZ in the positive and negative modes is shown in Figure 2C,D. Each dot of the PCA model represents a plasma sample. The PCA score plot showed that the three groups of plasma samples were clearly divided into three categories, which indicated that the endogenous metabolites were obviously changed. The OPLS-DA model results of NC/ABS and ABS/EZ groups in the positive and negative modes are shown in Figure 3. The potential metabolic markers in the NC, ABS, and EZ groups were analyzed. The parameters shown in Table 1 indicated that the predictive capability of the models was excellent. An S-plot of the OPLS-DA model shown in Figure 3 was structured to confirm the potential metabolic markers. In this S-plot, every point represents an ion *m*/*z*-Rt pair. The X axis represents the contribution of the variables, while the Y axis represents the confidence of the variables. The farther the data point from the center, the greater the contribution of this point to the separation of the two groups. Therefore, the points at either end of the S-shaped curve represent potential Q-markers with the highest contribution [37].

### 2.3. Identification of Potential Endogenous Metabolite Markers

The variables (*m*/*z*-Rt pairs) that contributed to the distinction were filtered by the variable importance in projection (VIP) value > 1.0 of the OPLS-DA model and *p* value < 0.05 of *t*-test. After that, the possible chemical structures of these variables needed to be identified by the human metabolome database (HMDB) (http://www.hmdb.ca) within a high-accuracy quasi-molecular ion and a mass error of 5 ppm. Moreover, the initially identified metabolites were compared with the MS/MS spectrum to further determine the structure of the compound. The potential endogenous metabolite markers identified are shown in Table 2.

### 2.4. Metabolic Pathway Analysis

MetaboAnalyst 4.0 (http://www.metaboanalyst.ca), a powerful tool of metabolic pathway analysis, was used to analyze the metabolic pathway of the potential endogenous metabolite markers. The method was as follows: Put the HMDB ID of the potential endogenous metabolite markers into MetaboAnalyst 4.0, choose Rattus norvegicus (rat) as the pathway library, and set the hypergeometric test as the over representation analysis method, and relative-betweeness centrality as the pathway topology analysis method. A summary of the pathway analysis with MetPA is shown in Figure 4. The x-axis (pathway impact) represents the importance of the metabolic pathway. The y-axis (−logP) represents the significance of the metabolic pathway enrichment analysis. The values of pathway impact and –logP were larger, the correlation of the metabolic difference between different groups was higher, and the circle in Figure 4 became bigger and redder. In this study, the pathway impact-value was set to 0.10 according to reference [37]. The pathway analysis results with MetaboAnalyst 4.0 are shown in Table 3. The results indicated that the disturbed pathways of acute blood stasis and EZhu (EZ) treatment were linoleic acid metabolism, ether lipid metabolism, sphingolipid metabolism, glycerophospholipid metabolism, glutathione metabolism, arachidonic acid metabolism, glyoxylate and dicarboxylate metabolism, pentose and glucuronate interconversions, tryptophan metabolism, glycerolipid metabolism, steroid hormone biosynthesis, and lysine degradation. The metabolic pathway map included in the therapeutic effects of EZ on ABS is shown in Figure 5. To entirely and visually exhibit the connections and differences among different groups, the relative intensity of the 26 differential metabolites in plasma was displayed with a heatmap generated by MeV4.9.0 software. The results are shown in Figure 6.

### 2.5. Component-Target-Pathway Network Analysis

The component-target-pathway network results are shown in Figure 7. The network predicted that 124 targets were closely related to 21 chemical components, in which 68 targets were closely related to blood stasis. It also predicted that 616 metabolic pathways were related to 68 targets, in which five metabolic pathways, including linoleic acid metabolism, arachidonic acid metabolism, glycerolipid metabolism, sphingolipid metabolism, and steroid hormone biosynthesis, corresponded with plasma metabolomics results.

## 3. Discussion

### 3.1. Model Evaluation

Based on the theory of TCM, blood stasis is originally caused by anger emotions and external environmental factors, such as a cold condition, which is mainly induced by abnormal blood flow and viscosity. The BSS model established in this study is based just on this theory [38]. Hemorheology is a science that studies the rule of blood flow and deformation. Coagulation is a process involving the enzymatic activation of a series of plasma coagulation factors. At present, clinical practice mainly use four blood coagulation indexes, including PT, APTT, TT, and FIB to reflect the status of the coagulation system in body. TT reflects the common coagulation pathway in plasma. APTT indicates the changes in the intrinsic coagulation pathway. PT indicates the changes in the extrinsic coagulation pathway [39]. FIB represents the content of fibrinogen. Hemorheology and four blood coagulation indexes are the primary indexes to evaluate the blood stasis diseases. These parameters make a great contribution to the study of the effect mechanism of blood stasis. In this study, EZ had a good effect on hemorheology and four blood coagulation indexes in ABS model rats. In addition, modern studies show that blood stasis is closely related to the fibrinolytic system and inflammatory factors. Tissue-type plasminogen activator (t-PA) can promote fibrinolysis, while plasminogen activator inhibitor-1 (PAI-1) can reduce fibrinolytic function and promote blood stasis in peripheral microcirculation and increase the tendency of thrombosis [40]. If t-PA and PAI-1 are out of balance, it will accelerate the formation of blood stasis syndrome. Our current research showed that EZ could improve the imbalance of fibrinolytic factors caused by blood stasis. Meanwhile, EZ could also reduce the level of inflammatory factors of IL-6 and α-TNF induced by blood stasis.

### 3.2. Metabolic Pathway and Function Analysis

Metabolic pathway analysis results showed that blood stasis was mainly related to lipid metabolism and amino acid metabolism, which were in accordance with the previous reports [41,42]. Metabolic pathways associated with lipid metabolism, including linoleic acid metabolism, ether lipid metabolism, sphingolipid metabolism, glycerophospholipid metabolism, and arachidonic acid metabolism. LysoPCs were resulted from partial hydrolysis of phosphatidylcholines, which removes one of the fatty acid groups. The hydrolysis is generally the result of the enzymatic action of phospholipase A2. LysoPCs were important metabolites in lipid metabolism, which also play an important part in the development of many diseases, including atherosclerosis, diabetes, cancer, inflammation, and blood lipid disorders [43,44]. In the progress of coagulation, LysoPC can restrain the transcriptional activity of tissue factors and NF-κB by disrupting the binding site of κB, thereby raising cAMP levels of monocytes to regulate the expression of tissue factors that take part in thrombosis [45]. Compared with the NC group, LysoPCs, including LysoPC (16:0), PC (18:1(9Z)e/2:0), LysoPC (O-18:0), PE (16:1(9Z)/22:1(13Z), Phosphorylcholine, PE (P-16:0e/16:0), and PC (16:1(9Z)/18:1(9Z)) in the ABS group were all significantly declined in the ABS group and increased in the EZ group, indicating that glycerophospholipid metabolism plays an important role in the formation and cure of blood stasis syndrome.

Arachidonic acid, an n-6 polyunsaturated fatty acids, has a vital function in inflammatory progress, and can work as a substrate for the production of some pro-inflammatory eicosanoids, leading to the production of inflammatory mediators, such as tumor necrosis factor alpha (TNF-α) and interleukin-1 (IL-1) [46]. Moreover, thromboxane A2 (TXA2), a product of arachidonic acid, plays a key role in thrombosis. The pathological imbalance of TXA2 may motivate platelet aggregation, vasospasm, and thrombosis, which can lead to angina pectoris, myocardial infarction, and cerebrovascular accident [47]. Therefore, we speculate that the pro-inflammatory effect of arachidonic acid may be one of the contributing factors for the formation of blood stasis syndrome. In the present study, arachidonic acid and TXA2 in the ABS group were obviously increased compared with NC group and reversed by EZ preadministration. These results were in agreement with previous studies and the pharmacological results of this study. Linoleic acid, another n-6 polyunsaturated fatty acid, can be metabolized to arachidonic acid, which plays an important role in inflammatory processes. Mensink’s meta-analysis, which summarized 60 controlled feeding studies, discovered that polyunsaturated fatty acids (primarily linoleic acid), when substituted for carbohydrates, decrease low-density lipoprotein cholesterol and slightly increase high-density lipoprotein cholesterol, and thus decrease the total cholesterol to high-density lipoprotein ratio, a summary indicator that predicts cardiovascular disease risk [48]. Compared with the NC group, the level of linoleic acid was significantly down-regulated in the ABS group and reversed by EZ preadministration.

Sphingolipids are bioactive constituents of cell membranes, which play a crucial part in cell growth, differentiation, senescence, and apoptosis. Sphingolipids primary cycle as a portion of lipoproteins in the blood. Ceramides and glycosphingolipids are the main metabolites of the sphingolipid metabolic pathway. Previous studies showed that both ceramides and glycosphingolipids participated in the regulation of vascular growth and vascular tone, which might be the possible mechanism of these putative biochemical modulators of hypertension and atherosclerosis [49]. In this study, metabolites in sphingolipid metabolism, including ceramide, dihydroceramide, glucosylceramide, sphingomyelin, and sulfatide, were all returned to the normal levels by EZ preadministration.

In addition, we also discovered that amino acid metabolism was obviously disturbed by acute blood stasis. Kynurenine is an important producer of l-tryptophan due to its immune-suppressive action. Evidence proved that kynurenine could accelerate activated Th cells apoptosis, induce immunosuppressive Treg cells differentiation, and restrain immunoreactions [50,51]. In this study, l-tryptophan was reversed from the ABS group to the EZ group. Except the above metabolic pathways, metabolites in glutathione metabolism, glyoxylate and dicarboxylate metabolism, pentose and glucuronate interconversions, steroid hormone biosynthesis, and lysine degradation were also recovered to the level of the NC group.

## 4. Materials and Methods

### 4.1. Chemicals and Materials

LC-MS grade acetonitrile, LC-MS grade methyl alcohol, HPLC grade methanoic acid, (Merck. Co. Inc., Darmstadt, Germany), and ultra-pure grade water obtained from a Milli-Q system (Millipore, Bedford, MA, USA). The other solvents were of analytical grade.

The fresh rhizome of *Curcuma wenyujin* was collected from Rui-an, Zhejiang, China in December, 2016. The samples were identified as rhizome of *Curcuma wenyujin* by Professor Tulin Lu at Nanjing University of Chinese Medicine. The fresh rhizome was steamed for 1.5 h, then sliced in 3 mm thickness, and oven dried at 60 °C to process into EZhu according to the Chinese Pharmacopoeia Pharmacopeia 2015 Edition.

The Elisa kits of four coagulation function parameters, including prothrombin time (PT), activated partial thrombin time (APTT), thromboplastin time (TT), and fibrinogen (FIB), were purchased from Beijing Steellex Scientific Instrument Company (Beijing, China). The Elisa kits of tissue-type plasminogen activator (t-PA), plasminogen activator inhibitor-1 (PAI-1), interleukin-6 (IL-6), and tumor necrosis factor-α (TNF-α) were purchased from Nanjing SenBeiJia Biological Technology Co., Ltd, Nanjing, China. Adrenaline hydrochloride injection was obtained from SuiCheng Pharmaceutical Co., Ltd. (Henan, Xinzheng, China). Compound salvia tablets were obtained from Yunnan Tongda biopharmaceutical Co., Ltd. (Yunnan, Dali, China).

### 4.2. Preparation of EZhu Sample

The water extract of EZhu was prepared as follows: 1 kg of EZhu was mixed with 10 times amount of distilled water and soaked for 0.5 h. Second, the mixture was boiled for 0.5 h and afterwards filtered by 4 layers of carbasus. This method was repeated to extract the residue. Afterwards, two water extracts were merged and vacuum-concentrated to 500 mL, equivalently 2 g/mL of crude drug, by a rotary evaporator at 55 °C. Furthermore, compound salvia tablets were melted with distilled water to 1 mg/mL. The samples were stored at 4 °C for future use.

### 4.3. Animals and Drug Administration

The acute blood stasis (ABS) rat model was established according to the literature [52,53,54,55,56]. A total of 32 specific pathogen free (SPF) degree male Sprague-Dawley (SD) rats (180 ± 20 g) were obtained from the animal breeding farm of Qinglong mountain in Jiangning District, Nanjing, China (license approval number: SYXK (Su) 2017-0001). Before the experiment, the rats were fed in the environment controlled breeding room. A 12 h light/dark cycle was set. Room temperature and relative humidity were regulated at 20 ± 2 °C and 55 ± 5 °C, respectively. In this study, all experimental animals were allowed free access to food and tab water. All rats were adapted for 7 days before the experiment. The study protocol was in accordance with the Guide for the Care and Use of Laboratory Animals, and was approved by the Animal Experimental Ethics Committee of Nanjing University of Chinese Medicine.

A total of 32 rats was randomly divided into 4 groups. Each group had 8 rats. Four groups were set as below: Normal control group (NC), acute blood stasis model group (ABS), EZhu group (EZ), and compound salvia tablets (CST) group, which was also the positive control group. EZ and CST were intragastrically administered to the EZ group (4.5 g/kg/day) and CST group (1.5 g/kg/day), respectively, for 7 days. The dose of administration was converted according to the clinical equivalent dose of rats. NC and ABS groups were intragastrically administered normal saline. The ABS models in ABS, CST, and EZ groups were established on the 7th day after dose according to the references: 1 h after the last potion, subcutaneous injected 0.1% adrenaline 0.8 mL/kg, 2 h later, the rats were made to swim in ice water (0–4 °C) for 5 min, another 2 h later, subcutaneous injection of 0.1% adrenaline 0.8 mL/kg for the second time. After that, all rats were intraperitoneal injected with 10% chloral hydrate. During this time, all rats were managed to fasting, but access to clean water. Blood samples were collected by the abdominal aortic method. 

### 4.4. Hemorheology and Related Functional Parameters

All the blood samples were collected by sodium citrate blood collection tubes (1:9). Off each rat, two tubes of blood sample were collected. One tube was used for testing blood viscosity and analyzed by a SA-5000 semiautomatic blood rheometer (Beijing Steellex Scientific Instrument Company, Beijing, China). The other tube of blood sample was centrifuged at 3000 r/min for 10 min to get an upper plasma sample for related function detection. APTT, PT, TT, FIB, t-PA, PAI-1, and TNF-α were carried out in perfect accordance with the product’s instructions. The remaining plasma samples were stored in a −80 °C refrigerator for metabolomics experiments subsequently.

### 4.5. Plasma Sample Preparation

Before mass spectrometric detection, the plasma samples were unfrozen at ambient temperature. Eight times methanol was added to the 200 μL plasma sample. The mixed samples were shaken violently for 30 s by a vortex mixer. Then, the mixture was centrifuged at 12,000 r/min at 4 °C for 10 min. The supernate was prepared for mass detection.

### 4.6. Mass Spectrum Analysis and Verification of Methodology

Plasma samples analysis, managed by Shimadzu UPLC (Kyoto, Japan), consisted of an LC-30AD Binary liquid pump, SIL-30SD auto sampler, DGU-20A5R On-Line Solvent Degasser, CTO-30A column oven, AB SCIEX Triple TOF 5600+ system, and ESI source. Chromatographic conditions were as follows: Agilent C18 reversed phase column (2.1 mm × 100 mm, 1.8 μm, Palo Alto, CA, USA), Mobile phase A (0.1% Formic acid aqueous solution)-B(acetonitrile), gradient elution program: 0–1 min, 5–25% B; 1–3 min, 25–30% B; 3–13 min, 30–55% B; 13–15 min, 55–70% B; 15–25 min, 70–100% B; 25–28 min, 100–5% B; flow rate: 0.3 mL/min; column temperature: 35 °C; injection volume: 1 μL. Mass spectrometer condition: ESI source, data collection in positive and negative ion mode, respectively. The source parameters were set as follows: Ion spray voltage floating: +4500/−4500; declustering potential: +60/−60 V; source temperature: 550 °C; the atomizing gas is N2, curtain gas: 35 psi; gas1(nebulizer gas): 55 psi; gas2(heater gas): 55 psi; collision energy: +35/−35e V; using MS/MS secondary mass spectrometry mode: The MS spectrometer ion scanning range was *m*/*z* 100–2000. The MS/MS spectrometer ion scanning range was *m*/*z* 50–1000; turn on dynamic background subtraction. The quality control (QC) sample was obtained by even mixing of all groups of plasma samples to ensure system suitability. Before the samples started testing, the QC sample was continuously detected 6 times. Moreover, system consistency was verified by QC samples after every 5 detected samples.

### 4.7. Multivariate Statistical Analysis

Analyst TF 1.6 software (AB Sciex, Boston, MA, USA) was used to extract the original metabolic fingerprint profiles. The MarkerView1.2.1 software (AB Sciex, Boston, MA, USA) was used for peak detection and alignment of the raw UPLC-Q/TOF-MS data. The parameters were set as follows: Minimum spectral peak width: 25 ppm; minimum RT peak width: 6 scans; noise threshold: 100; mass tolerance within 10 ppm; retention time tolerance within 0.5 min; and used area integrated from raw data. After being normalized to the total ion intensity per chromatogram, the original dates were translated into three-dimensional data matrices, including peak name (retention time-*m*/*z* value), sample ID, and normalized peak areas. Then, the three-dimensional data matrices were imported into the Simca-P14.1 software (Umetrics AB, Umea, Sweden) for chemometrics analysis, including PCA and OPLS-DA. PCA was applied to overview the metabolic profiles of plasma samples in different groups. OPLS-DA was used to discover endogenous biomarkers between different groups [57]. The predictive capability of the model was evaluated by the R2X (cum) and Q2X (cum) values in its score plot. The R2X (cum) indicates the explanatory capacity of the variables, while the Q2X (cum) value represents the predictive capability of the model. The method indicated good fitness when the R2X (cum) and Q2 (cum) values were both close to 1.0. A scatter plot (S-plot) of OPLS-DA was conducted to find the potential metabolic biomarkers among the NC, ABS, and EZ groups. Variable importance in projection (VIP) on behalf of the contribution degree of variables in different groups [58]. The potential metabolic biomarkers of the NC/ABS and ABS/EZ groups were evaluated according to VIP > 1.0 from OPLS-DA and *p* value < 0.05 from t-test. One-way analysis of variance (ANOVA) was managed by the Statistical Package for Social Science program (SPSS 20.0, Chicago, IL, USA) to evaluate the hemorheology, coagulation function, and other pharmacological indicators. The significance threshold was set at *p* < 0.05 for the test.

### 4.8. Network Target, Pathway Prediction of EZ Primary Components.

According to our previous study (shown in Appendix A), the 21 primary compounds in EZ were chosen to predict the biological targets. The canonical SMILES of 21 components were put into SwissTargetPrediction database (http://www.swisstargetprediction.ch/) to obtain the Uniprot ID of predict targets. Then, the Uniprot ID was imported into the Kyoto Encyclopedia of Genes and Genomes (KEGG) database (http://www.genome.jp/kegg/) to predict the related pathways. The biological targets related to blood stasis were selected from the GAD database (https://geneticassociationdb.nih.gov/). Then, the components-targets-pathways network was established by Cytoscape 3.6.1 software (Bethesda, MD, USA).

## 5. Conclusions

In summary, plasma metabolomics combined with the network pharmacology method provides a powerful approach to study the therapeutic mechanisms of *Curcuma wenyujin* rhizome (EZ) on blood stasis. The pharmacological results showed that EZ could effectively improve hemorheology, coagulation function, and other related indicators in blood stasis model rats. Furthermore, the metabolomics results indicated that 26 differential metabolites in lipid metabolism and the amino acid metabolism might be linked with EZ prevention in ABS. Five metabolic pathways predicted by network pharmacology were corresponded with plasma metabolomics results. Thus, our findings suggest that the LC-MS metabolomics method together with network pharmacology would be useful to explore its pathological mechanisms and clarify the mechanisms of action of EZ.

## Figures and Tables

**Figure 1 molecules-24-00082-f001:**
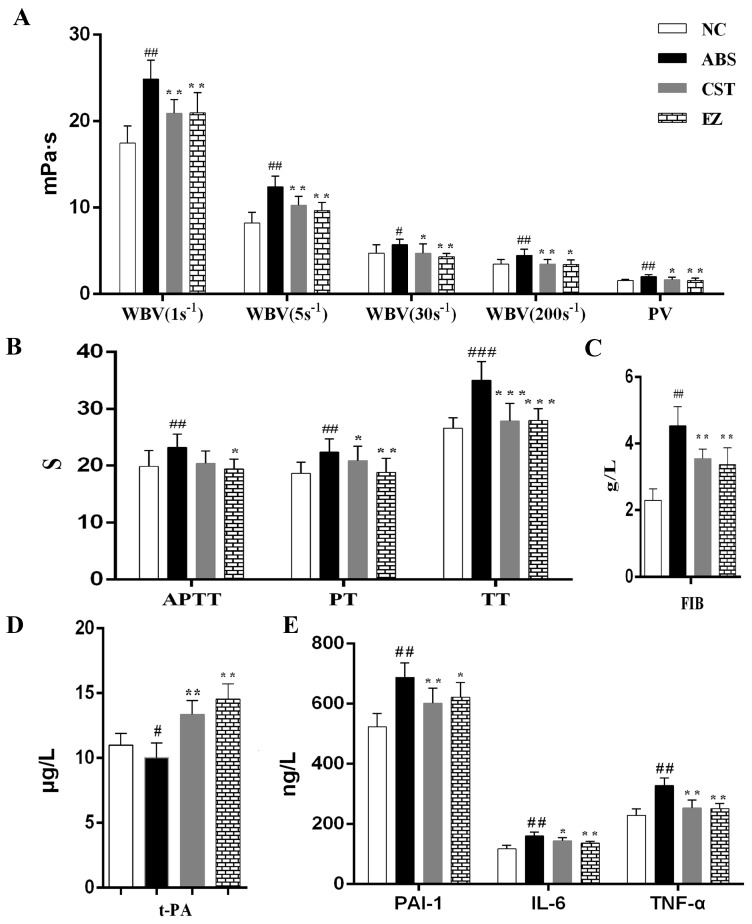
The influence of EZ on the related functions in normal control (NC), acute blood stasis (ABS), compound salvia tablets (CST), and EZ groups. Note: Whole blood viscosity, including 1 s^−1^, 5 s^−1^, 30 s^−1^, 200 s^−1^ shear rates, and plasma viscosity (**A**); four blood coagulation indexes, including PT, APTT, TT, and FIB (**B**,**C**); fibrinolytic factors and inflammatory factors (**D**,**E**); compared with NC group, ^###^, ^##^ and ^#^ represent *p* < 0.001, *p* < 0.01 and *p* < 0.05, respectively; compared with ABS group, ***, ** and * represent *p* < 0.001, *p* < 0.01 and *p* < 0.05, respectively.

**Figure 2 molecules-24-00082-f002:**
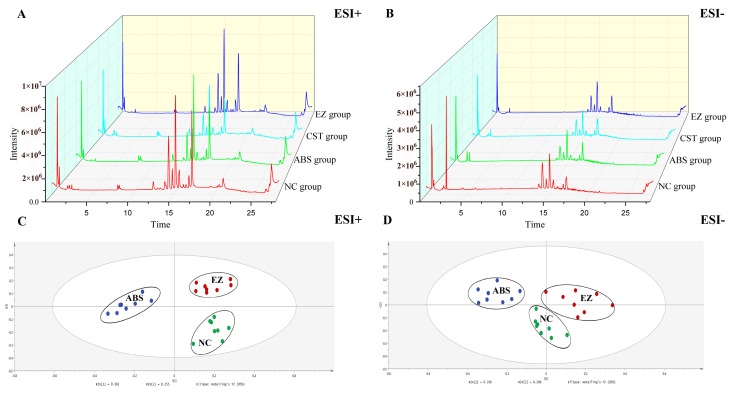
UPLC-Q/TOF-MS total ion chromatograms (TIC) of plasma samples among NC, ABS, CST, and EZ groups in ESI+ mode (**A**) and ESI− mode (**B**); PCA score plots of plasma metabolic profiling of NC, ABS, and EZ in ESI+ mode (**C**) and ESI− mode (**D**).

**Figure 3 molecules-24-00082-f003:**
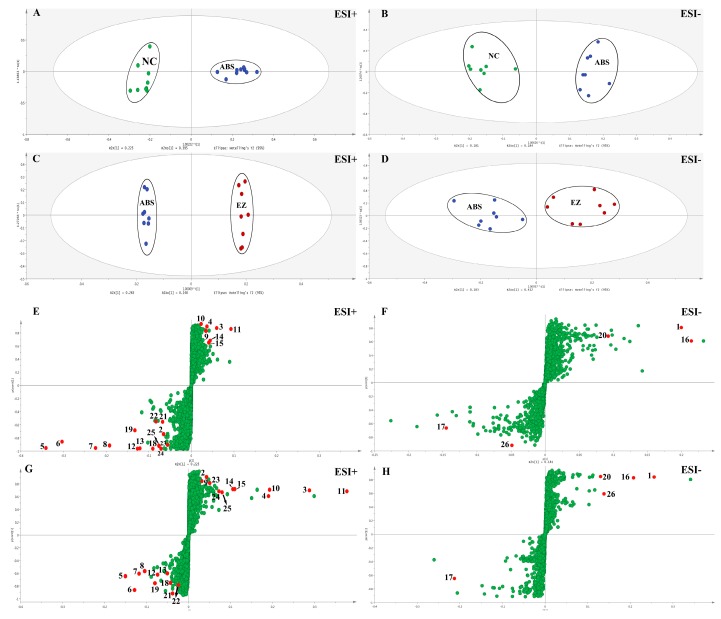
OPLS-DA model results of plasma metabolic profiling of NC/ABS and ABS/EZ in ESI+ mode (**A**,**C**) and ESI− mode (**B**,**D**); OPLS-DA s-plots of plasma metabolic profiling of NC/ABS and ABS/EZ in ESI+ mode (**E**,**G**) and ESI− mode (**F**,**H**). Note: The numbered red dots represent 26 potential biomarkers respectively, which are shown in Table 2.

**Figure 4 molecules-24-00082-f004:**
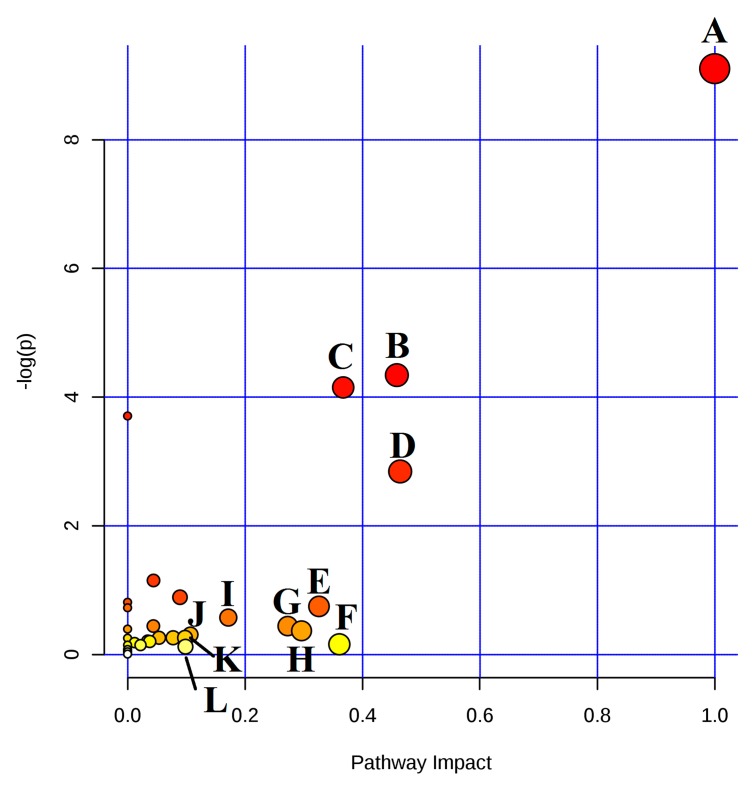
Summary of pathway analysis with MetPA. (**A**) Linoleic acid metabolism; (**B**) sphingolipid metabolism; (**C**) glycerophospholipid metabolism; (**D**) ether lipid metabolism; (**E**) arachidonic acid metabolism; (**F**) glutathione metabolism; (**G**) pentose and glucuronate interconversions; (**H**) glyoxylate and dicarboxylate metabolism; (**I**) tryptophan metabolism; (**J**) glycerolipid metabolism; (**K**) lysine degradation; (**L**) steroid hormone biosynthesis.

**Figure 5 molecules-24-00082-f005:**
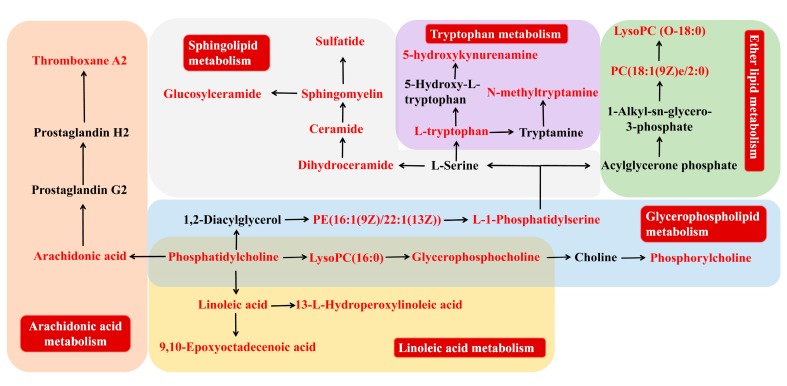
Metabolic networks of potential metabolite markers in response to the preventive effect of EZ for acute blood stasis. The identified metabolite markers are highlighted in red.

**Figure 6 molecules-24-00082-f006:**
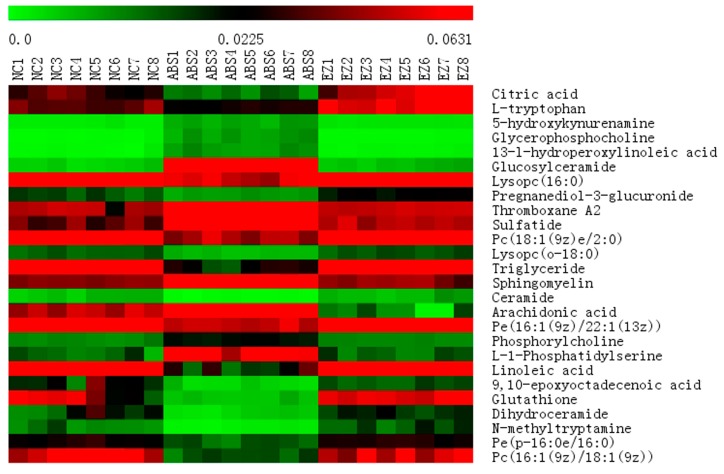
Heat map of differential metabolites.

**Figure 7 molecules-24-00082-f007:**
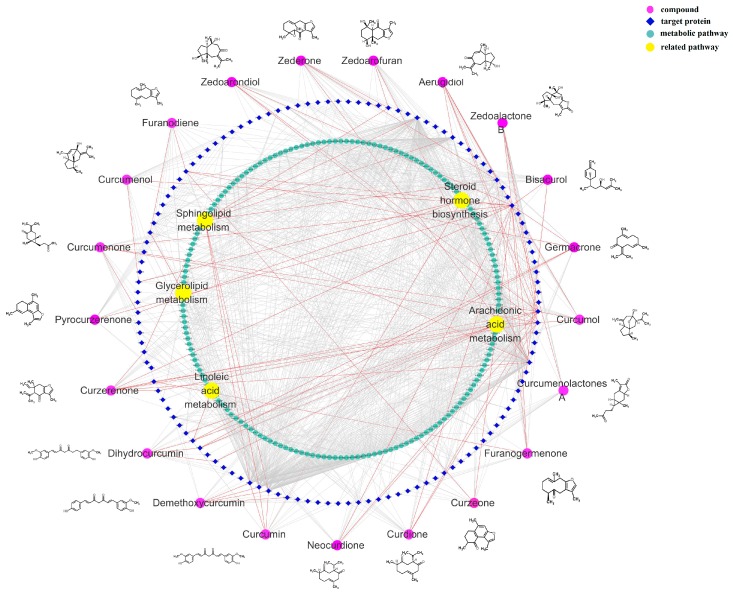
Compound-target-metabolic pathway network.

**Table 1 molecules-24-00082-t001:** Parameters of PCA and OPLS-DA models.

No.	Model	ESI+			ESI−		
		R^2^X	R^2^Y	Q^2^	R^2^X	R^2^Y	Q^2^
M1	PCA	0.731	-	0.423	0.706	-	0.403
M2	OPLS-DA	0.772	0.994	0.957	0.569	0.956	0.864
M3	OPLS-DA	0.687	0.998	0.880	0.616	0.843	0.734

Note: M1 on behalf of the PCA model of metabolic profiling of NC, ABS, and EZ groups; M2 on behalf of the OPLS-DA model of metabolic profiling of NC and ABS groups; M3 on behalf of the OPLS-DA model result of metabolic profiling of ABS and EZ groups.

**Table 2 molecules-24-00082-t002:** Potential biomarkers in plasma associated with ABS based on the UPLC-Q-TOF/MS analysis.

No.	*m*/*z*	Rt	VIP	HMDB ID	Compound Name	Formula	ABS	EZ	Mode	Delta (ppm)
1	173.0090	2.12	2.22	HMDB0000094	Citric acid	C6H8O7	↓_*_	↑#	−	2
2	205.0969	2.12	2.26	HMDB0000929	l-tryptophan	C11H12N2O2	↓_*_	↑#	+	1
3	163.0879	7.26	1.18	HMDB0004076	5-hydroxykynurenamine	C9H12N2O2	↑_*_	↓#	+	4
4	275.1372	7.83	1.04	HMDB0000086	Glycerophosphocholine	C8H20NO6P	↑_*_	↓#	+	2
5	330.2638	10.02	1.10	HMDB0003871	13-l-hydroperoxylinoleic acid	C18H32O4	↑_**_	↓#	+	0
6	426.3573	15.06	1.13	HMDB0004975	Glucosylceramide	C48H91NO8	↑_**_	↓##	+	1
7	496.3400	15.32	20.05	HMDB0010382	LysoPC (16:0)	C24H50NO7P	↓_*_	↑##	+	0
8	497.6450	15.32	1.21	HMDB0010318	Pregnanediol-3-glucuronide	C27H44O8	↓_**_	↑##	+	4
9	338.3410	16.89	5.32	HMDB0003208	Thromboxane A2	C20H40O	↑_*_	↓#	+	2
10	806.5437	17.29	1.97	HMDB0012317	Sulfatide	C42H79NO11S	↑_*_	↓#	+	1
11	550.3879	17.67	2.65	HMDB0011148	PC (18:1(9Z) e/2:0)	C28H56NO7P	↓_*_	↑##	+	2
12	510.3922	17.98	1.32	HMDB0011149	LysoPC (O-18:0)	C26H56NO6P	↓_**_	↑##	+	1
13	297.2791	19.06	4.46	HMDB0005368	Triglyceride	C57H108O6	↓_**_	↑#	+	1
14	703.5755	20.48	1.76	HMDB0010169	Sphingomyelin	C39H80N2O6P	↑_*_	↓#	+	0
15	359.3345	20.53	1.16	HMDB0004951	Ceramide	C38H75NO3	↓_*_	↑#	+	0
16	303.2339	20.91	1.55	HMDB0001043	Arachidonic acid	C20H32O2	↑_**_	↓##	−	3
17	830.5951	20.92	1.53	HMDB0008974	PE (16:1(9Z)/22:1(13Z))	C43H82NO8P	↓_*_	↑##	−	4
18	184.0741	21.08	1.12	HMDB0001565	Phosphorylcholine	C5H15NO4P	↑_*_	↓#	+	1
19	784.5901	21.11	2.07	HMDB0010164	l-1-Phosphatidylserine	C44H86NO10P	↑_*_	↓##	+	5
20	279.2342	21.31	1.57	HMDB0000673	Linoleic acid	C18H32O2	↓_*_	↑#	−	4
21	357.3004	21.62	2.54	HMDB0004701	9,10-epoxyoctadecenoic acid	C18H32O3	↓_*_	↑#	+	0
22	371.1007	24.29	2.36	HMDB0000125	Glutathione	C10H17N3O6S	↓_*_	↑#	+	3
23	352.2838	24.99	2.17	HMDB0006752	Dihydroceramide	C19H39NO3	↓_*_	↑#	+	5
24	366.2652	25.09	1.39	HMDB0004370	*N*-methyltryptamine	C11H14N2	↓_**_	↑##	+	0
25	759.5736	26.79	1.11	HMDB0011158	PE(P-16:0e/16:0)	C37H74NO7P	↓_*_	↑##	+	2
26	802.5635	26.79	1.69	HMDB0008005	PC (16:1(9Z)/18:1(9Z))	C42H80NO8P	↓_*_	↑##	−	2

Note: ↓, content decreased; ↑, content increased. *****, *p* < 0.05, compared with NC group, **, *p* < 0.01, compared with NC group; #, *p* < 0.05, compared with ABS group, ##, *p* < 0.01, compared with ABS group.

**Table 3 molecules-24-00082-t003:** Pathway analysis result with MetaboAnalyst 4.0.

No.	Pathway Name	Hits	Total	Raw p	−log(p)	Impact
1	Linoleic acid metabolism	4	5	1.11 × 10^−4^	9.1056	1
2	Ether lipid metabolism	3	13	0.058157	2.8446	0.46429
3	Sphingolipid metabolism	5	21	0.013011	4.342	0.45865
4	Glycerophospholipid metabolism	6	30	0.015752	4.1508	0.36728
5	Glutathione metabolism	1	26	0.85366	0.15822	0.36069
6	Arachidonic acid metabolism	3	36	0.47359	0.74741	0.32601
7	Glyoxylate and dicarboxylate metabolism	1	16	0.69218	0.36791	0.2963
8	Pentose and glucuronate interconversions	1	14	0.64306	0.44152	0.27273
9	Tryptophan metabolism	3	41	0.56347	0.57364	0.17157
10	Glycerolipid metabolism	1	18	0.7346	0.30843	0.10704
11	Steroid hormone biosynthesis	3	70	0.88595	0.1211	0.09862
12	Lysine degradation	1	20	0.77122	0.25978	0.09783

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
