# Peer review of "Mechanism of *Curcuma wenyujin* Rhizoma on Acute Blood Stasis in Rats Based on a UPLC-Q/TOF-MS Metabolomics and Network Approach"

_molecules, 2018, doi:10.3390/molecules24010082_

Round 1
Reviewer 1 Report
Considering your paper reported about mechanism on acute blood stasis in rats, my minor comment on this study is that the manuscript does not provide a convincing case regarding the really new contributions of metabolomics.
Author Response
Q: Considering your paper reported about mechanism on acute blood stasis in rats, my minor comment on this study is that the manuscript does not provide a convincing case regarding the really new contributions of metabolomics.
Response:Thank you for your suggestions and comments on our manuscript entitled “Mechanism of Curcuma wenyujin rhizoma on acute blood stasis in rats based on a UPLC-Q/TOF-MS metabolomics and network approach” (molecules-402926). These comments are very valuable and helpful for revising and improving our paper. Revised portion are marked in “Track Changes” function in the revised manuscript. The main corrections in the paper and the responds to the reviewer’s comments are as following:
Firstly, we have noticed that you suggested us to improve the introduction to provide sufficient research background. According to this advice, we have added more progress in research on blood stasis, the main chemical components of Curcuma Wenyujin rhizome and related references in the introduction section to make the research background more sufficient. The content was added from Line 31 to 43, 48 to 53.
Secondly, your primary evaluation of this paper was focused on the new contribution of metabolomics. Actually, in this study, we have established an acute blood stasis model on rats. The hemorheology, coagulation functions and other related pharmacodynamic indexes were significantly changed (P<0.05) in model group which indicate the acute blood stasis model was successfully built. The related pharmacodynamic indexes were significantly improved by Curcuma Wenyujin rhizome preadministration which demonstrated the curative effect of Curcuma Wenyujin rhizome. During this treatment, we have found significant changes in related endogenous metabolites by metabolomics study. Then the possible pathological mechanism of blood stasis was presumed and verified by pharmacodynamic test indicators, according to the changes of these endogenous metabolites. For example, the significant changes of linoleic acid, arachidonic acid showed that the occurrence of blood stasis is related to inflammation which is accordance with abnormities of IL-6 and TNF-α tested in this study. This result is consistent with the literature report. Moreover,the changes of metabolites in sphingolipid metabolism including ceramide, dihydroceramide, glucosylceramide, sphingomyelin, and sulfatide indicate blood stasis is associated with vascular growth and vascular tone. In addition to speculation about the possible mechanism of blood stasis, this research is the first time to elucidate the therapeutic mechanism of Curcuma Wenyujin rhizome in blood stasis from metabolic perspective. In a word, this paper can provide a research basis for further study on the underlying mechanism of Curcuma wenyujin rhizome in the treatment of blood stasis and a new example of metabolomics in the study of TCM mechanisms. Thank you very much for your careful review.
Reviewer 2 Report
The manuscript attempts to use metabolomics to examine a condition described by TCM as Blood stasis syndrome. From the introduction, it seems to be related to cardiovascular diseases. There are only two references (37 and 38) describing the animal model. 37 appears to be a chinese paper. In ref 38, it seems that the model is built by injecting adrenaline hydrochloride into the animal after placing in ice-cold water. Ref 38 mentioned that injection of adenaline hydrochloride and exposure to ice-cold water might induce blood stasis, which is ok. However, exposing the animal to such a stressful condition might not just induce blood statis alone! Adenaline has many physicological effects, pubchem suggested that it "causes systemic VASOCONSTRICTION and gastrointestinal relaxation, stimulates the HEART, and dilates BRONCHI and cerebral vessels. It is used in ASTHMA and CARDIAC FAILURE and to delay absorption of local ANESTHETICS." Given the extensive physicological effects of adenaline, I am not surprised to see a large changes in the endogenous metabolites, however, do that relate to Blood Stasis Syndrome remains uncertain. The authors need to justify or claify this in the paper.
Author Response
Q: The manuscript attempts to use metabolomics to examine a condition described by TCM as Blood stasis syndrome. From the introduction, it seems to be related to cardiovascular diseases. There are only two references (37 and 38) describing the animal model. 37 appears to be a chinese paper. In ref 38, it seems that the model is built by injecting adrenaline hydrochloride into the animal after placing in ice-cold water. Ref 38 mentioned that injection of adenaline hydrochloride and exposure to ice-cold water might induce blood stasis, which is ok. However, exposing the animal to such a stressful condition might not just induce blood statis alone! Adenaline has many physicological effects, pubchem suggested that it "causes systemic VASOCONSTRICTION and gastrointestinal relaxation, stimulates the HEART, and dilates BRONCHI and cerebral vessels. It is used in ASTHMA and CARDIAC FAILURE and to delay absorption of local ANESTHETICS." Given the extensive physicological effects of adenaline, I am not surprised to see a large change in the endogenous metabolites, however, do that relate to Blood Stasis Syndrome remains uncertain. The authors need to justify or claify this in the paper.
Response: Thank you for your suggestions and comments on our manuscript entitled “Mechanism of Curcuma wenyujin rhizoma on acute blood stasis in rats based on a UPLC-Q/TOF-MS metabolomics and network approach” (molecules-402926). These comments are very valuable and helpful for revising and improving our paper. Revised portion are marked in “Track Changes” function in the revised manuscript. The main corrections in the paper and the responds to the reviewer’s comments are as following:
We noticed that your evaluation of this paper was focused on the animal model we established in this study. Actually, at the beginning of this research, we have read a lot of literatures on blood stasis model. The related literatures have been added in the revised manuscript. Meanwhile, we have added interpretation in the discussion section that “Based on the theory of TCM, blood stasis is originally caused by anger emotions and external environmental factors such as a cold condition which is mainly induced by abnormal blood flow and viscosity. The BSS model which established in this study is based just on this theory” [28]. The content was added from Line 206 to 208. In addition to references, an important indicator of the success of a pathological animal model is the clinical evaluation index. Hemorheology and coagulation functions were important indexes to evaluate the blood stasis in Clinic. In this study, hemorheology including whole blood viscosity and plasma viscosity and four coagulation indeses including PT, APTT, TT and FIB were tested and significantly changed after twice subcutaneous injections of adrenaline hydrochloride and exposed to ice-cold water. Meanwhile, these indexes were significantly reversed by preadministration of Curcuma wenyujin rhizoma. Based on the above reasons, we considered that the acute blood model we established is successful and applicable to this research. Thank you for your careful review.
Reviewer 3 Report
In this study, the authors investigated the effect of Curcuma wenyujin rhizome on acute blood stasis and its underlying mechanism using a metabolomic approach. The therapeutic effects of Curcuma wenyujin rhizome against blood stasis was confirmed by its regulation of blood coagulation indexes, fibrinolytic and inflammatory factors. The important metabolites and their related metabolomic pathways regulated by Curcuma wenyujin rhizome were revealed by UPLC-QTOF MS fingerprinting and multivariate analysis including PCA and OPLS-DA. Overall the experiments were well designed and conducted, experimental data were properly interpreted. This study highlights the therapeutic potential of Curcuma wenyujin rhizome on treatment of blood stasis conditions and add insights on its potential mechanism of action.
Some concerns:
1. “Curcuma wenyujin” (italic font) should be used across the manuscript.
2. Some sentences need to be rewritten in materials and methods. e.g. line 290, “Collected the blood from the aortaventralis”; line 298, “Stored the remaining plasma samples…”; line 306, “…..Shimadzu UPLC (Japan) coupled with an LC-30AD Binary liquid pump….”. (better to use “consists of” than “coupled with” for UPLC components, except mass spec).
3. Fig. 1 can be presented in different sections (A,B,C etc.), and the explanation of different sections need to be added in figure caption. Also, explain “***” and “###” in figure.
4. Fig. 2, figure quality (resolution etc.) need to be improved. Add figure section (A-D) in results.
5. Fig. 3, figure quality also need to be improved. For S-plots, consider marking the selected biomarkers (Table 2) in different color.
6. The chemical profile of Curcuma wenyujin rhizome was not explored in the study. A LC-MS analysis on extracted used on animal study can be valuable if major phytochemicals can be identified.
Author Response
Dear reviewer:
Thank you for your suggestions and comments on our manuscript entitled “Mechanism of Curcuma wenyujin rhizoma on acute blood stasis in rats based on a UPLC-Q/TOF-MS metabolomics and network approach” (molecules-402926). These comments are very valuable and helpful for revising and improving our paper. Revised portion are marked in “Track Changes” function in the revised manuscript. The main corrections in the paper and the responds to the reviewer’s comments are as follows:
Q 1: “Curcuma wenyujin” (italic font) should be used across the manuscript.
Response 1: We have changed all the font of “Curcuma wenyujin” into italic font throughout the manuscript and marked in “Track Changes” function in the revised manuscript. Thank you very much for pointing out this mistake.
Q 2: Some sentences need to be rewritten in materials and methods. e.g. line 290, “Collected the blood from the aortaventralis”; line 298, “Stored the remaining plasma samples…”; line 306, “…. Shimadzu UPLC (Japan) coupled with an LC-30AD Binary liquid pump….”. (better to use “consists of” than “coupled with” for UPLC components, except mass spec).
Response 2: We have changed the sentences which you suggested and checked the English language and style by native English staff at our university.
Q 3: Fig. 1 can be presented in different sections (A, B, C etc.), and the explanation of different sections need to be added in figure caption. Also, explain “***” and “###” in figure.
Response 3: We have presented Fig 1 in A-E sections and added the explanation of different sections in figure caption. The explanation of “***” and “###” were also added in figure caption. Thank you very much for your careful inspection.
Q 4: Fig. 2, figure quality (resolution etc.) need to be improved. Add figure section (A-D) in results.
Response 4: We have improved the resolution of Fig.2 from 300 dpi to 600 dpi and added the figure sections (A-D) in results. Fig.2 has been uploaded as an attachment separately for the word software will automatically compress the inserted figure. Thank you for your suggestion.
Q 5: Fig. 3, figure quality also needs to be improved. For S-plots, consider marking the selected biomarkers (Table 2) in different color.
Response 5: We have improved the resolution of Fig.3 from 300 dpi to 600 dpi and marking the selected biomarkers in red color of S-plot. Fig.3 has been uploaded as an attachment separately.
Q 6: The chemical profile of Curcuma wenyujin rhizome was not explored in the study. A LC-MS analysis on extracted used on animal study can be valuable if major phytochemicals can be identified.
Response 6: Actually, in the previous study, the chemical profile of Curcuma Wenyujin rhizome was analyzed by our researcher. We have put the total ion chromatogram of Curcuma Wenyujin rhizome by UPLC-Q/TOF-MS (Fig S1.) and identification of 21 chemical compounds (Table S1.) in supplementary materials. Due to we have explained too little about this research background, so you haven’t notice it. You reminded us that we should introduce the pharmacological action of the main chemical composition in Curcuma Wenyujin rhizome. We have added these contents in the introduction section. The new content was put from Line 49 to 52. Thank you very much for your useful advice.
Round 2
Reviewer 2 Report
This version gave slightly more introduction on the Acute Blood Stasis syndrome, however, I think this part still need to be strengthen as Molecules is not a journal focusing on Traditional Medicines, I would guess most readers would require a more introduction. It is important to justify the animal model is a good representation of "Blood Stasis". Ref 3-5 was cited to correlate "Blood Stasis" with the some quantifiable end-points. However reference 3 seems not to be the primary source, but it cited a conference paper. I think this may not be a good reference as no detail can be further found. Ref 4 and 5 seems to be chinese papers (with English abstract available). Ref 4 is based on human studies and I am interested to see how does the criteria 1-3 in ref 4 correlate with the parameters that the authors are measuring in this study. TCM is based on clinical experiences in the past, thus if ref 4 is based on the TCM diagnostic and identify some measurable endpoints which can be mimic in the animal model, then the manuscript would be more convincing. Perhaps, the author may also explain a little bit why they did not focus on a disease caused by Blood Stasis? Blood Stasis may be difficult to quantify and study, From the data present, I am not fully convinced with the last statement of the conclusion.
Author Response
Q: This version gave slightly more introduction on the Acute Blood Stasis syndrome. however, I think this part still need to be strengthen as Molecules is not a journal focusing on Traditional Medicines, I would guess most readers would require a more introduction. It is important to justify the animal model is a good representation of "Blood Stasis". Ref 3-5 was cited to correlate "Blood Stasis" with some quantifiable end-points. However, reference 3 seems not to be the primary source, but it cited a conference paper. I think this may not be a good reference as no detail can be further found. Ref 4 and 5 seems to be chinese papers (with English abstract available). Ref 4 is based on human studies and I am interested to see how does the criteria 1-3 in ref 4 correlate with the parameters that the authors are measuring in this study. TCM is based on clinical experiences in the past, thus if ref 4 is based on the TCM diagnostic and identify some measurable endpoints which can be mimic in the animal model, then the manuscript would be more convincing. Perhaps, the author may also explain a little bit why they did not focus on a disease caused by Blood Stasis? Blood Stasis may be difficult to quantify and study, From the data present, I am not fully convinced with the last statement of the conclusion.
Response: Thank you for your careful review and useful comments on our manuscript entitled “Mechanism of Curcuma wenyujin rhizoma on acute blood stasis in rats based on a UPLC-Q/TOF-MS metabolomics and network approach” (molecules-402926). These comments are very valuable and helpful for revising and improving our paper. Moreover, it can be seen that you have a very in-depth study of TCM, according to your suggestions. Therefore, I have read a lot of literatures related to clinical diagnosis of blood stasis to improve our paper and have deepened my understanding of blood stasis. Revised portion are marked in “Track Changes” function in the revised manuscript. The main corrections in the paper and the responds to the reviewer’s comments are as follows:
Q 1: This version gave slightly more introduction on the Acute Blood Stasis syndrome. however, I think this part still need to be strengthen as Molecules is not a journal focusing on Traditional Medicines, I would guess most readers would require a more introduction.
Response 1: I have added more traditional Chinese medicine (TCM) theory of blood stasis in the introduction section to strengthen my research background. I mainly supplemented the concept of Zheng in TCM and the relationship among Zheng, disease and TCM therapy. In TCM, Zheng is not merely a subclass disease but also a type of common symptom discovery in different diseases. The doctors of TCM often recognize Zheng by identifying little difference in the same symptoms of the same disease. Therefore, the distinction of Zheng makes TCM therapy become individualized in some degree [12]. In addition, patients with specific similar symptoms of different diseases could be treated by the same TCM treatment according to the theory of TCM, which has been accepted by the Consolidated Standards of Reporting Trials (CONSORT) for Chinese Herbal Medicine Formulas 2017 [13]. It also explains why this study choose a blood stasis model rather than a disease model related to blood stasis. The content was added from Line 39 to 45.
Q 2: It is important to justify the animal model is a good representation of "Blood Stasis". Ref 3-5 was cited to correlate "Blood Stasis" with some quantifiable end-points. However, reference 3 seems not to be the primary source, but it cited a conference paper. I think this may not be a good reference as no detail can be further found.
Response 2: I’m sorry for my carelessness to cited a conference paper in my study. I have changed Ref 3 to Ref 2-5 in the new revised manuscript. Ref 2 is a review article which elaborated the possible aetiopathogenesis of BSS from the perspective of senescence of red blood cells (RBCs). The accumulation of senescent RBCs and their products induce pathological conditions that affect blood flow resistance and cause thrombosis, vasoconstriction and methemoglobinemia. Ref 3-4 are original articles which have established acute blood stasis and tested hemorheology and other related indicators. Ref 5 is also a review article which had acquired experts' opinions on the definition and diagnosis of blood stasis in order to establish a modern concept of blood stasis. Six main categories: (1) blood stasis concepts; (2) blood stasis-related biomarkers; (3) methods of diagnosing blood stasis; (4) drugs for promoting blood circulation and dissipating stasis; (5) blood stasis-related diseases; and (6) blood stasis-related societies were including in this paper.
Q 3: Ref 4 and 5 seems to be chinese papers (with English abstract available). Ref 4 is based on human studies and I am interested to see how does the criteria 1-3 in ref 4 correlate with the parameters that the authors are measuring in this study. TCM is based on clinical experiences in the past, thus if ref 4 is based on the TCM diagnostic and identify some measurable endpoints which can be mimic in the animal model, then the manuscript would be more convincing.
Response 3: Ref 4 is just an English abstract. After careful inspection, I found out that both Ref 4 and 5 are not to be good references. I have changed Ref 4-5 to Ref 6 in the new revised manuscript. Ref 6 have detailed the diagnosis of BSS which was based on a grading system drafted by the committees of the International Conference on Blood Stasis Syndrome (October, 1988, Beijing, China). In this reference, whole blood viscosity and plasma viscosity were two of the diadynamic criteria of BSS. Meanwhile, hemorheology and four blood coagulation indexes were tending to be the end points of BSS in the review article and research article of blood stasis which I cited in this paper. About the combination of TCM diagnostic criteria with test indicators of BSS, I think the abnormalities of hemorheology and four blood coagulation indexes can associate with tingling sensation in a fixed position, blood spots under the skin, and purplish tongue or petechiae on the tongue. It is easy to understand that increased blood viscosity will causes slower blood flow, stagnant blood, then further lead to tingling sensation in a fixed position. The abnormalities of four blood coagulation indexes will stimulate the coagulation system and lead to blood spots, purplish tongue or petechiae on the tongue. Such speculation remains to be scientifically verified. In fact, it has always been a difficult point to combine the clinical diagnosis of TCM with the evaluation indexes of Western medicine in the basic research of TCM, and it is also the direction we need to work for.
Q 4: Perhaps, the author may also explain a little bit why they did not focus on a disease caused by Blood Stasis? Blood Stasis may be difficult to quantify and study, From the data present, I am not fully convinced with the last statement of the conclusion.
Response 4: Actually, I have answered this question in Response 1. The reason why I choose Blood Stasis rather than a disease caused by Blood Stasis is Zheng has an important position in TCM theory. It (Zheng) is the basis for TCM doctors to diagnose diseases and work out treatment plans. Different disease manifestations may be due to the same Zheng. That’s why we focus on Blood Stasis, though it is difficult to quantify and study.
References
2. You, S.; Park, B. Accelerated RBC senescence as a novel pathologic mechanism of blood stasis syndrome in traditional East Asian medicine. Am J Transl Res. 2015, 7(3), 422-429.
3. Wang, Y.; Yan, J. Composition of The Essential Oil From Danggui-zhiqiao Herb-Pair and Its Analgesic Activity and Effect on Hemorheology in Rats With Blood Stasis Syndrome. Pharmacogn Mag. 2016, 12(48), 271-275.
4. Ning, S.Y.; Jiang, B.P. Effect of Liangxuehuayu Recipe on hemorheology in rats with blood stasis syndrome. Asian Pac J Trop Med. 2012, 5(12), 935-938.
5. Choi, T.Y.; Jun, J.H. Expert opinions on the concept of blood stasis in China: An interview study. Chin J Integr Med. 2016, 22(11), 823-831.
6. Liao, J.; Liu, Y. Identification of more objective biomarkers for Blood-Stasis syndrome diagnosis. BMC Complement Altern Med. 2016, 16(1), 371.
12. He, H.; Chen, G. Xue-Fu-Zhu-Yu capsule in the treatment of qi stagnation and blood stasis syndrome: a study protocol for a randomized controlled pilot and feasibility trial. Trials. 2018, 19(1), 515.
13. Cheng, C.W.; Wu, T.X. CONSORT extension for Chinese herbal medicine formulas 2017: recommendations, explanation, and elaboration. Ann Intern Med. 2017, 167(2), 112-121.